# NEWTON LOSSES: EFFICIENTLY INCLUDING SECOND-ORDER INFORMATION INTO GRADIENT DESCENT

## ABSTRACT

We present Newton losses, a method for incorporating second-order information of losses by approximating them with quadratic functions. The presented method is applied only to the loss function and allows training the neural network with gradient descent. As loss functions are usually substantially cheaper to compute than the neural network, Newton losses can be used at a relatively small additional cost. We find that they yield superior performance, especially when applied to non-convex and hard-to-optimize loss functions such as algorithmic losses, which have been popularized in recent research.

## 1 INTRODUCTION

Neural network training has gained a tremendous amount of attention in machine learning in recent years. This is primarily due to the success of backpropagation and stochastic gradient descent for first-order optimization. However, there has also been a strong line of work on second-order optimization for neural network training; see [1] and references therein. While these second-order optimization methods (such as Newton's method and natural gradient descent) exhibit improved convergence rates and therefore require fewer training steps, they have two major limitations [2], namely (i) computing the curvature (or its inverse) for a large and deep neural network is computationally substantially more expensive than simply computing the gradient with backpropagation, which makes second-order methods practically inapplicable in most cases; (ii) networks trained with second-order information exhibit reduced generalization capabilities [3].

In this work, we propose a novel method for incorporating second-order information of the loss function into training, while training the actual neural network with gradient descent. As loss functions are usually substantially cheaper to evaluate than a neural network, the idea is to apply second-order optimization to the loss function while training the actual neural network with first-order optimization. For this, we decompose the original iterative optimization problem into a two-stage iterative optimization problem, which leads to Newton losses. This is especially interesting for intrinsically hard-to-optimize loss functions, i.e., where second-order optimization of the inputs to the loss is superior to first-order optimization. Such loss functions have recently become increasingly popular, as they allow for solving more specialized tasks such as inverse rendering [4]–[6], learning-to-rank [7]–[13], self-supervised learning [14], differentiation of optimizers [15], [16], and top-k supervision [9], [11], [17]. In this paper, we summarize these loss functions exceeding typical classification and regression under the umbrella of algorithmic losses [18] as they introduce algorithmic knowledge into the training objective.

We evaluate the proposed Newton losses for various algorithmic losses on two popular benchmarks: the four-digit MNIST sorting benchmark and the Warcraft shortest-path benchmark. We found that Newton losses improve the performance in case of hard-to-optimize losses and maintain the original performance in the case of easy-to-optimize losses.

**Contributions.** The contributions of this work are (i) introducing a mathematical framework for splitting iterative optimization methods into two-stage schemes, which we show to be equal to the original optimization methods; (ii) introducing Newton losses as combinations of first-order and second-order optimization methods; and (iii) an extensive empirical evaluation on two algorithmic supervision benchmarks using an array of algorithmic losses.

## 1.1 Related Work

The related work comprises algorithmic supervision losses and second-order optimization methods. To the best of our knowledge, this is the first work combining second-order optimization of loss functions with first-order optimization of neural networks, especially for algorithmic losses.

**Algorithmic Losses.** Algorithmic losses, i.e., losses which contain some kind of algorithmic component, have become quite popular in recent machine learning research. In the domain of recommender systems, early learning-to-rank works already appeared in the 2000s [7], [8], [19], but more recently [20] propose differentiable ranking metrics, and [12] propose PiRank, which relies on differentiable sorting. For differentiable sorting an array of methods has been proposed in recent years, which includes NeuralSort [21], SoftSort [22], Optimal Transport Sort [9], differentiable sorting networks (DSN) [11], and the relaxed Bubble Sort algorithm [18]. Other works explore differentiable sorting-based top-k for applications such as differentiable image patch selection [23], differentiable k-nearest-neighbor [17], [21], top-k attention for machine translation [17], and differentiable beam search methods [17], [24]. But algorithmic losses are not limited to sorting: other works have considered learning shortest-paths [15], [16], [18], learning 3D shapes from images and silhouettes [4]–[6], [18], [25], [26], learning with combinatorial solvers for NP-hard problems [16], learning to classify handwritten characters based on editing distances between strings [18], learning with differentiable physics simulations [27], and learning protein structure with a differentiable simulator [28], among many others. The methods used to make algorithms differentiable can be broadly categorized into those, which estimate gradients via sampling (e.g., [15]) and those, which have analytical closed-form gradient estimates (e.g., [18]). In this work, we specifically focus on the tasks of ranking supervision and shortest-path supervision, and discuss the methods that we consider in detail in Section 4.

**Second-Order Optimization.** Second-order methods have recently gained popularity in machine learning due to their fast convergence properties when compared to first-order methods [1]. One alternative to vanilla Newton are quasi-Newton methods, which, instead of computing an inverse Hessian in the Newton step (which is expensive), approximate this curvature from the change in gradients [2], [29], [30]. In addition, a number of new approximations to the pre-conditioning matrix have been proposed in the recent literature, i.a., [31]–[33]. While the vanilla Newton method relies on the Hessian, there are variants which use the empirical Fisher information matrix, which can coincide in specific cases with the Hessian, but generally exhibits somewhat different behavior. For an overview and discussion of Fisher-based methods (sometimes referred to as natural gradient descent), see [34], [35].

**Decomposition Methods.** When working with complex and non-standard loss functions, the optimization problem for neural network training is often challenging. In these cases, a natural approach is to look for a decomposition, i.e., to break up the optimization problem into two (or more) sub-problems, which are then solved sequentially. The idea of decomposing an optimization problem is not new, see [36] for an overview of decomposition methods. A decomposition method, which has become particularly popular in machine learning, is the alternating direction method of multipliers [37]. There are other decomposition methods known as operator splitting methods [38], which include the methods of multipliers with Gauss-Seidel passes [37], coordinate descent-type methods with linear coupling constraints [2], or consensus based optimization schemes [39], [40]. These methods typically derive optimization problems via Lagrangian duality that are then solved sequentially and to the respective (global) optimality which requires loss functions that can be optimized at relatively low cost. Moreover, convergence guarantees can be obtained if the underlying optimization problem is convex and if the coupling constraints are linear. In this work, we consider problems where neither of these properties hold.

## 2 A Two-Stage Optimization Method

We consider the training of a neural network $f(x; \theta)$, where $x \in \mathbb{R}^n$ is the vector of inputs, $\theta \in \mathbb{R}^d$ is the vector of weights and $y = f(x; \theta) \in \mathbb{R}^m$ is the vector of outputs. We assume that we have access to a data set of $N$ samples drawn from the input distribution, which describes the empirical input $\mathbf{x} = [x_1, \ldots, x_N]^\top \in \mathbb{R}^{N \times n}$. As per vectorization, we denote $\mathbf{y} = f(\mathbf{x}; \theta) \in \mathbb{R}^{N \times m}$ as the matrix describing the outputs of the neural network corresponding to the empirical inputs. Further, let $\ell : \mathbb{R}^{N \times m} \to \mathbb{R}$ denote the loss function, and let the ground truth output be implicitly encoded in $\ell$ (because for many algorithmic losses, it is not simply a label, but could be ordinal information).

In a general setting, the training of a neural network can be expressed as the optimization problem

$$\arg\min_{\theta\in\Theta}\ell(f(\mathbf{x};\theta))\,, \tag{1}$$

where $\Theta\subseteq\mathbb{R}^d$ is the domain of the parameters $\theta$, and $f$ and $\ell$ are such that the minimum in (1) exists. Note that the formulation (1) is extremely general and includes, e.g., optimization of non-decomposable loss functions (i.e., not composed of individual losses per training sample), which is relevant for some algorithmic losses like ranking losses.

Typically, the optimization problem (1) is solved by using some iterative algorithm like gradient descent (or Newton's method) updating the weights $\theta$ by repeatedly applying the following step:

$$\theta\leftarrow\text{One optimization step of }\ell(f(\mathbf{x};\theta))\text{ wrt. }\theta\,. \tag{2}$$

However, in this work, we consider decomposing the optimization problem (1) into two problems, which may be solved by applying the following two updates in an alternating fashion:

$$\mathbf{z}^{\star}\leftarrow\text{One optimization step of }\ell(\mathbf{z})\text{ wrt. }\mathbf{z}=f(\mathbf{x};\theta)\,, \tag{3a}$$

$$\theta\leftarrow\text{One optimization step of }\tfrac{1}{2}\|\mathbf{z}^{\star}-f(\mathbf{x};\theta)\|_2^2\text{ wrt. }\theta\,. \tag{3b}$$

This split allows us later to use two different iterative optimization algorithms for (3a) and (3b), respectively. This is especially interesting for optimization problems where the loss function $\ell$ is non-convex and its minimization is a difficult optimization problem itself, and as such those problems, where a stronger optimization method exhibits a superior rate of convergence.

We can express individual optimization steps (corresponding to (2) and (3a)) via

$$\theta\leftarrow\arg\min_{\theta'\in\Theta}\ell(f(\mathbf{x};\theta'))+\Omega(\theta',\theta)\qquad\text{and}\qquad\mathbf{z}^{\star}\leftarrow\arg\min_{\mathbf{z}\in\mathcal{Y}}\ell(\mathbf{z})+\Omega(\mathbf{z},f(\mathbf{x};\theta)) \tag{4}$$

where $\Omega$ is a regularizer such that one step of a respective optimization method corresponds to the global optimum of the regularized optimization problems in (4). The regularizer $\Omega$ has the standard property that $\Omega(a,b)=0$ for any $a=b$. Note that the explicit form of the regularizer $\Omega$ does not need to be known. Nevertheless, in Supplementary Material C, we discuss explicit choices of $\Omega$.

This allows us to express the set of points of convergence for the iterative optimization methods. Recall that an iterative optimization method has converged if it has arrived at a fixed point, i.e, the parameters do not change when applying an update. The set of points of convergence for (2) is

$$\mathcal{A}=\left\{\theta\mid\theta\in\arg\min_{\theta'}\ell(f(\mathbf{x};\theta'))+\Omega(\theta',\theta)\right\}, \tag{5}$$

i.e., those points at which the update does not change $\theta$. For the two-stage optimization method (3), the set of points of convergence is

$$\mathcal{B}=\left\{\theta\mid f(\mathbf{x};\theta)=\mathbf{z}^{\star}\in\arg\min_{\mathbf{z}}\ell(\mathbf{z})+\Omega(\mathbf{z},f(\mathbf{x};\theta))\right\} \tag{6}$$

as the method has converged if the update (3a) yields $\mathbf{z}^{\star}=\mathbf{z}$ because the subsequent update (3b) will not change $\theta$ as $\mathbf{z}^{\star}=\mathbf{z}=f(\mathbf{x};\theta)$ already holds, and thus $\tfrac{1}{2}\|\mathbf{z}^{\star}-f(\mathbf{x};\theta)\|_2^2=0$. Now, we show that the iterative method (2) and the alternating method (3) lead to the same sets of convergence points.

**Lemma 1** (Equality of the Sets of Convergence Points). *The set $\mathcal{A}$ of points of convergence obtained by the iterative optimization method* (2) *is equal to the set $\mathcal{B}$ of points of convergence obtained by the two-step iterative optimization method* (3).

*Proof.* ($\mathcal{A}\subset\mathcal{B}$)  First, we show that any point in $\mathcal{A}$ also lies in $\mathcal{B}$. By definition, for each point in $\mathcal{A}$, the optimization step (2) does not change $\theta$, i.e., $\theta'=\theta$. Thus, $f(\mathbf{x};\theta)=f(\mathbf{x};\theta')\in\arg\min_{\mathbf{z}}\ell(\mathbf{z})+\Omega(\mathbf{z},f(\mathbf{x};\theta))$, and therefore $\theta\in\mathcal{B}$.

($\mathcal{B}\subset\mathcal{A}$)  Second, we show that any point in $\mathcal{B}$ also lies in $\mathcal{A}$. For each $\theta\in\mathcal{B}$, we know that, by definition, $f(\mathbf{x};\theta)=\mathbf{z}^{\star}\in\arg\min_{\mathbf{z}}\ell(\mathbf{z})+\Omega(\mathbf{z},f(\mathbf{x};\theta))$, therefore $\theta\in\arg\min_{\theta'}\ell(f(\mathbf{x};\theta'))+\Omega(f(\mathbf{x};\theta'),f(\mathbf{x};\theta))$ where $\Omega(f(\mathbf{x};\theta),f(\mathbf{x};\theta))=0=\Omega(\theta,\theta)$, and, therefore $\theta\in\mathcal{A}$. $\qquad\square$

While Lemma 1 states the equivalence of the original training (2) and its counterpart (3) wrt. their possible points of convergence (i.e., solutions) for an arbitrary choice of the iterative method, the two approaches are also equal when applying standard first-order or second-order optimization schemes. In other words, running a gradient descent step according to (2) coincides with two gradient steps of the alternating scheme (3a) and (3b), namely one step for (3a) and one step for (3b).

**Theorem 2** (Gradient Descent Step Equality between (2) and (3a)+(3b)). *A gradient descent step according to (2) with arbitrary step size $\eta$ coincides with two gradient descent steps, one according to (3a) and one according to (3b), where the optimization over $\theta$ has a step size of $\eta$ and the optimization over $z$ has a unit step size. Proof deferred to SM A.*

**Theorem 3** (Newton Step Equality between (2) and (3a)+(3b) for $m = 1$). *In the case of $m = 1$, a Newton step according to (2) with arbitrary step size $\eta$ coincides with two Newton steps, one according to (3a) and one according to (3b), where the optimization over $\theta$ has a step size of $\eta$ and the optimization over $z$ has a unit step size. Proof deferred to SM A.*

## 3 NEWTON LOSSES

In this section, we focus on the two-stage optimization method (3), and propose optimizing (3a) with Newton's method, while optimizing (3b) with stochastic gradient descent. Let us begin by considering the quadratic approximation of the loss function at the location $\mathbf{y} = f(\mathbf{x}; \theta)$, i.e.,

$$\tilde{\ell}_{\mathbf{y}}(\mathbf{z}) = \ell(\mathbf{y}) + (\mathbf{z} - \mathbf{y})^\top \nabla_{\mathbf{y}} \ell(\mathbf{y}) + \tfrac{1}{2}(\mathbf{z} - \mathbf{y})^\top \nabla_{\mathbf{y}}^2 \ell(\mathbf{y})(\mathbf{z} - \mathbf{y}) \,. \tag{7}$$

To find the location $\mathbf{z}^\star$ of the minimum of $\tilde{\ell}_{\mathbf{y}}(\mathbf{z})$, we set its derivative to $0$:

$$\nabla_{\mathbf{z}} \tilde{\ell}_{\mathbf{y}}(\mathbf{z}^\star) = 0 \quad \Leftrightarrow \quad \nabla_{\mathbf{y}} \ell(\mathbf{y}) + \nabla_{\mathbf{y}}^2 \ell(\mathbf{y})(\mathbf{z}^\star - \mathbf{y}) = 0 \tag{8}$$

$$\Leftrightarrow \quad \nabla_{\mathbf{y}} \ell(\mathbf{y}) = -\nabla_{\mathbf{y}}^2 \ell(\mathbf{y})(\mathbf{z}^\star - \mathbf{y}) \quad \Leftrightarrow \quad -\left(\nabla_{\mathbf{y}}^2 \ell(\mathbf{y})\right)^{-1} \nabla_{\mathbf{y}} \ell(\mathbf{y}) = \mathbf{z}^\star - \mathbf{y} \tag{9}$$

Thus, the minimum of $\tilde{\ell}_{\mathbf{y}}(\mathbf{z})$ is $\mathbf{z}^\star = \arg\min_{\mathbf{z}} \tilde{\ell}_{\mathbf{y}}(\mathbf{z}) = \mathbf{y} - (\nabla_{\mathbf{y}}^2 \ell(\mathbf{y}))^{-1} \nabla_{\mathbf{y}} \ell(\mathbf{y})$.

When $\ell$ is quadratic, it can be readily seen that $\mathbf{z}^\star$ is independent of the choice of $\mathbf{y}$. For non-quadratic functions, we heuristically assume independence as $\mathbf{z}^\star$ is the projected optimum / goal of the function $\ell$. In implementations, this independence can be achieved using `.detach()` or `.stop_gradient()`. Using $\mathbf{z}^\star$, we can derive the Newton loss $\ell^*$ as

$$\ell_{\mathbf{z}^\star}^*(\mathbf{y}) = \tfrac{1}{2}(\mathbf{z}^\star - \mathbf{y})^\top(\mathbf{z}^\star - \mathbf{y}) = \tfrac{1}{2} \|\mathbf{z}^\star - \mathbf{y}\|_2^2 \qquad \text{where} \qquad \mathbf{z}^\star = \arg\min_{\mathbf{z}} \tilde{\ell}_{\mathbf{y}}(\mathbf{z}) \tag{10}$$

and its derivative as $\nabla_{\mathbf{y}} \ell_{\mathbf{z}^\star}^*(\mathbf{y}) = \mathbf{y} - \mathbf{z}^\star$. With this construction, we obtain the Newton loss $\ell_{\mathbf{z}^\star}^*$, a new convex loss, which has a gradient that corresponds to the Newton step of the original loss.

Note that (10) is an instance of (3) for a regularization term describing the quadratic approximation error, i.e., $\Omega(\mathbf{z}, f(\mathbf{x}; \theta)) = \tilde{\ell}_{f(\mathbf{x};\theta)}(\mathbf{z}) - \ell(\mathbf{z})$. As $\Omega$ is already implicitly part of the Newton step, we do not need to evaluate it.

In general, $\ell_{\mathbf{z}^\star}^*$ exhibits more desirable behavior than $\ell$, as a single gradient descent step can solve any quadratic problem, and it possesses the same convergence properties as the Newton method when optimizing $\mathbf{y}$. In the case of non-convex $\ell$, as it is common for many algorithmic losses, the incorporation of second-order information also substantially improves empirical performance. Note that, in the case of non-convex or ill-conditioned settings, using Tikhonov regularization [41] stabilizes $\ell_{\mathbf{z}^\star}^*$.

In the following, we define Newton losses and use $x$ and $z^\star$ to denote samples / rows of $\mathbf{x}$ and $\mathbf{z}^\star$.

**Definition 1** (Element-wise Hessian-based Newton losses). *For a loss function $\ell$, and a given current parameter vector $\theta$, we define the element-wise Hessian-based Newton loss as*

$$\ell_{z^\star}^*(y) = \frac{1}{2} \|z_E^\star - y\|_2^2 \,, \quad where \quad z_E^\star = \bar{y} - (\nabla_{\bar{y}}^2 \ell(\bar{y}))^{-1} \nabla_{\bar{y}} \ell(\bar{y}) \quad and \quad \bar{y} = f(x; \theta) \,.$$

However, instead of using the element-wise Hessian-based Newton loss, it is typically more stable to use the empirical Hessian-based Newton loss.

**Definition 2** (Empirical Hessian-based Newton losses). *For a loss function $\ell$ and a given current parameter vector $\theta$, we define the empirical Hessian-based Newton loss as*

$$\ell_{z^\star}^*(y) = \frac{1}{2} \|z_H^\star - y\|_2^2 \,, \quad where \quad z_H^\star = \bar{y} - \left(\mathbb{E}_{\bar{y}}\left[\nabla_{\bar{y}}^2 \ell(\bar{y})\right]\right)^{-1} \nabla_{\bar{y}} \ell(\bar{y}) \quad and \quad \bar{y} = f(x; \theta) \,.$$

Instead of using the Hessian, it is also possible to use the Fisher information matrix for second-order information. While this coincides with the Hessian in certain cases, in most cases it yields different results. The Fisher-based Newton loss can be seen as using natural gradient descent for optimizing the loss, while using regular gradient descent for optimizing the neural network.

**Definition 3** (Fisher-based Newton losses). *For a loss function $\ell$, and a given current parameter vector $\theta$, we define the Fisher-based Newton loss as*

$$\ell_{z^\star}^*(y) = \frac{1}{2}\|z_F^\star - y\|_2^2, \quad \text{where} \quad z_F^\star = \bar{y} - \left(\mathbb{E}_{\bar{y}}\left[\nabla_{\bar{y}}\ell(\bar{y})\nabla_{\bar{y}}\ell(\bar{y})^\top\right]\right)^{-1}\nabla_{\bar{y}}\ell(\bar{y}) \ \text{and} \ \bar{y} = f(x;\theta).$$

**Remark 1** (Computational Considerations). *The Hessian of the loss function $\nabla_y^2\ell(y)$ may be approximated using the empirical Fisher matrix $F = \mathbb{E}_y\left[\nabla_y\ell(y)\nabla_y\ell(y)^\top\right]$. However, as only the Hessian of the loss function (and not the Hessian of the neural network) needs to be computed, computing the exact Hessian $\nabla_y^2\ell(y)$ is usually also fast.*

**Remark 2** (Derivative of the Newton Loss). *The derivative of the Newton loss is*

$$\frac{\partial}{\partial y}\frac{1}{2}\|z^\star - y\|_2^2 = y - z^\star. \tag{11}$$

*Note that the derivative of $z^\star$ wrt. $y$ is zero because it is the projected optimum of the original loss.*

## 3.1 EXAMPLES

We have seen in (10) how a given loss function $\ell$ induces a corresponding Newton loss $\ell^*$. For specific loss functions, the Newton loss can be explicitly computed. We begin with the trivial example of the MSE loss. For notational simplicity we often drop the subscript $z^\star$ in the definition of the Newton loss (10).

**Example 1** (MSE loss). *Consider the classical MSE loss, i.e., $\ell(y) = \frac{1}{2}\|y - y^\star\|_2^2$, where $y^\star$ denotes the ground truth. Then, $z^\star = y^\star$ and accordingly the Newton loss is given as*

$$\ell_{z^\star}^*(y) = \frac{1}{2}\|z^\star - y\|_2^2 = \frac{1}{2}\|y^\star - y\|_2^2 = \ell(y).$$

*Therefore, the MSE loss $\ell$ and its induced Newton loss $\ell^*$ are equivalent.*

A popular loss function for classification is the softmax cross entropy (SMCE) loss, defined as

$$\ell_{\text{SMCE}}(y) = \sum_{i=1}^k -p_i\log q_i, \qquad \text{where} \qquad q_i = \frac{\exp(y_i)}{\sum_{j=1}^k\exp(y_j)}. \tag{12}$$

**Example 2** (Softmax cross-entropy loss). *For the SMCE loss, the induced Newton loss is given as*

$$\ell_{\text{SMCE}}^*(y) = \frac{1}{2}\|z^\star - y\|_2^2 \tag{13}$$

*where the element-wise Hessian variant is $z_E^\star = -\left(\text{diag}(q) - qq^\top\right)^{-1}(q - p) + y$,*

*the empirical Hessian variant is $z_H^\star = -\mathbb{E}_q\left[\text{diag}(q) - qq^\top\right]^{-1}(q - p) + y$,*

*and the empirical Fisher variant is $z_F^\star = -\mathbb{E}_q\left[(q - p)(q - p)^\top\right]^{-1}(q - p) + y$.*

In the experiments, we include a classification experiment with the SMCE loss. Additional examples of Newton losses can be found in the supplementary material.

## 4 ALGORITHMIC SUPERVISION LOSSES

In this section, we discuss how to derive the Newton losses for various types of algorithmic losses. Specifically, we consider SoftSort, DiffSort, AlgoVision, one-step Blackbox Differentiation, and stochastic smoothing. While all of these algorithmic losses can be used directly with Fisher-based Newton losses, the Hessian-based Newton losses require an estimation of the Hessian.

We consider the task of algorithmic supervision, i.e., problems where an algorithm is applied to the predictions of a model and only the outputs of the algorithm are supervised. Specifically, we focus on the tasks of ranking supervision and shortest-path supervision. As this requires backpropagating through conventionally non-differentiable algorithms, the respective approaches make the ranking or shortest-path algorithms differentiable such that they can be used as part of the loss function.

### 4.1 SoftSort and NeuralSort

SoftSort [22] and NeuralSort [21] are prominent yet simple examples of a differentiable algorithm. In the case of ranking supervision, they obtain an array or vector of scalars and return a row-stochastic matrix called the differentiable permutation matrix $P$, which is a relaxation of the argsort operator. Note that, in this case, a set of $k$ inputs yields a scalar for each image and thereby $y \in \mathbb{R}^k$. As a ground truth label, a ground truth permutation matrix $Q$ is given and the loss between $P$ and $Q$ is the binary cross entropy loss $\ell_{SS}(y) = \text{BCE}\left(P(y), Q\right)$. Minimizing the loss enforces the order of predictions $y$ to correspond to the true order, which is the training objective. SoftSort is defined as

$$P(y) = \text{softmax}\left(-\left|y^\top \ominus \text{sort}(y)\right|/\tau\right) = \text{softmax}\left(-\left|y^\top \ominus Sy\right|/\tau\right) \tag{14}$$

where $\tau$ is a temperature parameter, "sort" sorts the entries of a vector in non-ascending order, $\ominus$ is the element-wise broadcasting subtraction, $|\cdot|$ is the element-wise absolute value, and "softmax" is the row-wise softmax operator, as also used in (12) (right). NeuralSort is defined similarly and omitted for the sake of brevity. In the limit of $\tau \to 0$, SoftSort and NeuralSort converge to the exact ranking permutation matrix [21], [22]. A respective Newton loss can be implemented using automatic differentiation according to Definition 2 or via the Fisher information matrix using Definition 3.

### 4.2 DiffSort

Differentiable sorting networks (DSN) [11], [13] offer a strong alternative to SoftSort and NeuralSort. They are based on sorting networks, a classic family of sorting algorithms that operate by conditionally swapping elements [42]. As the locations of the conditional swaps are pre-defined, they are suitable for hardware implementations, which also makes them especially suited for continuous relaxation. By perturbing a conditional swap with a distribution and solving for the expectation under this perturbation in closed-form, we can differentially sort a set of values and obtain a differentiable doubly-stochastic permutation matrix $P$, which can be used via the BCE loss as in Section 4.1. We can obtain the respective Newton loss either via the Hessian computed via automatic differentiation or via the Fisher information matrix.

### 4.3 AlgoVision

AlgoVision [18] is a framework for continuously relaxing arbitrary simple algorithms by perturbing all accessed variables with logistic distributions. The method approximates the expectation value of the output of the algorithm in closed-form and does not require sampling. For shortest-path supervision, we use a relaxation of the Bellman-Ford algorithm [43], [44] and compare the predicted shortest path with the ground truth shortest path via an MSE loss. The input to the shortest path algorithm is a cost embedding matrix predicted by a neural network.

### 4.4 Stochastic Smoothing

Another differentiation method is stochastic smoothing [45]. This method regularizes a non-differentiable and discontinuous loss function $\ell(y)$ by randomly perturbing its input with random noise $\epsilon$ (i.e., $\ell(y + \epsilon)$). The loss function is then approximated as $\ell(y) \approx \ell_\epsilon(y) = \mathbb{E}_\epsilon[\ell(y + \epsilon)]$. While $\ell$ is not differentiable, its smoothed stochastic counterpart $\ell_\epsilon$ is differentiable and the corresponding gradient and Hessian can be estimated via the following result.

**Lemma 4** (Exponential Family Smoothing, adapted from Lemma 1.5 in Abernethy *et al.* [45])**.** *Given a distribution over $\mathbb{R}^m$ with a probability density function $\mu$ of the form $\mu(\epsilon) = \exp(-\nu(\epsilon))$ for any twice-differentiable $\nu$, then*

$$\nabla_y l_\epsilon(y) = \nabla_y \mathbb{E}_\epsilon\left[\ell(y + \epsilon)\right] = \mathbb{E}_\epsilon\left[\ell(y + \epsilon)\,\nabla_\epsilon \nu(\epsilon)\right], \tag{15}$$

$$\nabla_y^2 l_\epsilon(y) = \nabla_y^2 \mathbb{E}_\epsilon\left[\ell(y + \epsilon)\right] = \mathbb{E}_\epsilon\left[\ell(y + \epsilon)\left(\nabla_\epsilon \nu(\epsilon)\nabla_\epsilon \nu(\epsilon)^\top - \nabla_\epsilon^2 \nu(\epsilon)\right)\right]. \tag{16}$$

A *variance-reduced form* of (15) and (16) is

$$\nabla_y \mathbb{E}_\epsilon\left[\ell(y + \epsilon)\right] = \mathbb{E}_\epsilon\left[(\ell(y + \epsilon) - \ell(y))\,\nabla_\epsilon \nu(\epsilon)\right], \tag{17}$$

$$\nabla_y^2 \mathbb{E}_\epsilon\left[\ell(y + \epsilon)\right] = \mathbb{E}_\epsilon\left[(\ell(y + \epsilon) - \ell(y))\left(\nabla_\epsilon \nu(\epsilon)\nabla_\epsilon \nu(\epsilon)^\top - \nabla_\epsilon^2 \nu(\epsilon)\right)\right]. \tag{18}$$

In this work, we use this to estimate the gradient of the shortest path algorithm. By including the second derivative, we extend the perturbed optimizer losses to Newton losses. This also lends itself to full second-order optimization.

## 4.5 Perturbed Optimizers with Fenchel-Young Losses

Blondel *et al.* [46] build on stochastic smoothing and Fenchel-Young losses [47] to propose perturbed optimizers with Fenchel-Young losses. For this, they use algorithms, like Dijkstra, to solve optimization problems of the type $\max_{w \in \mathcal{C}} \langle y, w \rangle$, where $\mathcal{C}$ denotes the feasible set, e.g., the set of valid paths. Blondel *et al.* [46] identify the argmax to be the differential of max, which allows a simplification of stochastic smoothing. By identifying similarities to Fenchel-Young losses, they find that the gradient of their loss is

$$\nabla_y \ell(y) = \mathbb{E}_\epsilon \left[ \arg \max_{w \in \mathcal{C}} \langle y + \epsilon, w \rangle \right] - w^\star \tag{19}$$

where $w^\star$ is the ground truth solution of the optimization problem (e.g., shortest path). This formulation allows optimizing the model without the need for computing the actual value of the loss function. Blondel *et al.* [46] find that the number of samples—surprisingly—only has a small impact on performance, such that 3 samples were sufficient in many experiments, and in some cases even a single sample was sufficient. In this work, we confirm this behavior and also compare it to plain stochastic smoothing. We find that for perturbed optimizers, the number of samples barely impacts performance, while for stochastic smoothing more samples always improve performance. If only few samples can be afforded (like 10 or less), perturbed optimizers are better as they are more sample efficient; however, when more samples are available, stochastic smoothing is superior as it can utilize more samples better.

## 5 Experiments

For the experiments, we evaluate Newton losses on two applications of algorithmic supervision, i.e., problems where an algorithm is applied to the predictions of a model and the outputs of the algorithm are supervised. The first algorithmic supervision task is ranking supervision, where only the relative order of a set of samples is known, while their absolute values remain unsupervised. The second algorithmic supervision task is the shortest path supervision, where only the shortest path is supervised, while the underlying cost matrix remains unsupervised. Finally, as an ablation study, we also apply Newton losses to the trivial case of classification, where we do not expect a performance improvement, as the loss is not hard-to-optimize, but rather validate that the method still works.

## 5.1 Ranking Supervision

In this section, we explore the ranking supervision setting [13], [21] with an array of differentiable sorting-based losses. For this, we choose the recently popularized four-digit MNIST sorting benchmark [9], [11], [13], [18], [21], [22]. In this setting, sets of $n$ four-digit MNIST images are given, and the supervision is the relative order of these images corresponding to the displayed value, while the absolute values remain unsupervised. The goal is to learn a CNN that maps each image to a scalar value in an order preserving fashion. Since these losses are harder to optimize in this case, we can achieve substantial improvements over the baselines. We train the CNN using the Adam optimizer [48] at a learning rate of $10^{-3}$ for $100\,000$ steps and with a batch size of $100$.

Table 1: Differentiable sorting results. The metric is the percentage of rankings correctly identified (and individual element ranks correctly identified) averaged over 10 seeds.

| | $n = 5$ | | | | $n = 10$ | | | |
|---|---|---|---|---|---|---|---|---|
| | Cauchy DSN | Logistic DSN | NeuralSort | SoftSort | Cauchy DSN | Logistic DSN | NeuralSort | SoftSort |
| Regular | 85.09 (93.31) | 53.56 (77.04) | 71.33 (87.10) | 70.70 (86.75) | 55.29 (87.06) | 12.31 (58.81) | 24.26 (74.47) | 27.46 (76.02) |
| NL (Hessian) | **85.11 (93.31)** | **75.02 (88.53)** | 83.31 (92.54) | 83.87 (92.72) | **56.49 (87.44)** | **42.14 (75.35)** | **48.76 (84.83)** | **55.07 (86.89)** |
| NL (Fisher) | 84.95 (93.25) | 63.11 (79.28) | **83.93 (92.80)** | **84.03 (92.82)** | 56.12 (87.35) | 25.72 (52.18) | 39.23 (81.14) | 54.00 (86.56) |

We explore NeuralSort, SoftSort, and differentiable sorting networks (DSNs) with logistic and Cauchy distributions in Table 1. For NeuralSort and SoftSort, we find that using the Newton losses, based either on the Hessian or on the Fisher matrix, improve performance substantially. In this case, using the Hessian performs better than using the Fisher matrix. For DSNs, we find that for logistic DSNs, the improvements are substantial. Monotonic differentiable sorting networks, i.e., the Cauchy DSNs, provide an improved variant of differentiable sorting networks, which also have the property of quasi-convexity, and have been shown to exhibit much better training behavior. Thus, in this case, for $n = 5$ and Cauchy DSNs, there is no improvement over the default loss. However, for the somewhat harder setting of $n = 10$, we can observe an improvement of more than $1\%$ using the Hessian-based Newton loss, even in the Cauchy DSN case. In summary, we obtain strong improvements on losses that are difficult to optimize, while on well-behaving losses only small or no improvements can be achieved. This aligns with our goal of improving performance on losses that are hard-to-optimize.

## 5.2 Shortest-Path Supervision

In this section, we apply Newton losses to the shortest-path supervision task of the $12 \times 12$ Warcraft shortest-path problem [15], [16], [18]. Here, $12 \times 12$ Warcraft terrain maps are given as $96 \times 96$ RGB images, and the supervision is the shortest path from the top left to the bottom right according to a hidden cost embedding. The goal is to predict $12 \times 12$ cost embeddings of the terrain maps such that the shortest path according to the predicted embedding corresponds to the ground truth shortest path. For this task, we explore three approaches: the relaxed AlgoVision Bellman-Ford algorithm, stochastic smoothing, and perturbed optimizers.

For the relaxed AlgoVision Bellman-Ford algorithm, we explore two variants of the algorithm (an outer `For` loop and an outer `While` loop) and two losses ($L_1$ and $L_2^2$), i.e., a total of four settings. As computing the Hessian of the AlgoVision Bellman-Ford algorithm is too expensive with the PyTorch implementation, we restrict this case to the Fisher-based Newton loss. The variant with a `For` loop is easier to optimize than the `While` loop variant, as there is one condition less in the algorithm. This is

Table 2: Shortest-path benchmark results for different variants of the AlgoVision-relaxed Bellman-Ford algorithm. The displayed metric is the percentage of perfect matches averaged over 10 seeds.

| Algorithm Loop | For | | While | |
|---|---|---|---|---|
| Loss | $L_1$ | $L_2^2$ | $L_1$ | $L_2^2$ |
| Regular | 94.19 | **95.90** | 94.30 | 95.77 |
| Fisher Newton | **94.52** | 95.37 | **94.47** | **95.93** |

beneficial in case of regular training; however, the `For` loop variant yields lower quality shortest paths due to an artifact arising from many unnecessary additional loop traversals when backtracking was actually already finished. This is avoided in the `While` loop variant, which terminates the loop. As displayed in Table 2, the Newton loss improves performance in three out of four settings and the overall best performance is also provided by the Newton loss.

After discussing analytical relaxations, we continue with stochastic methods, the results of which are displayed in Table 3. For stochastic smoothing of the loss function, (i.e., stochastic smoothing applied to the algorithm and loss as one unit), we find that Newton losses improve the performance for 10 and 30 samples, while the regular

Table 3: Shortest-path benchmark results for the stochastic smoothing of the loss (including the algorithm), stochastic smoothing of the algorithm (excluding the loss), and perturbed optimizers with the Fenchel-Young loss. The metric is the percentage of perfect matches averaged over 10 seeds.

| Loss | SS of loss | | | SS of algorithm | | | PO w/ FY loss | | |
|---|---|---|---|---|---|---|---|---|---|
| # Samples | 3 | 10 | 30 | 3 | 10 | 30 | 3 | 10 | 30 |
| Regular | **62.83** | 77.01 | 85.48 | **57.55** | 78.70 | 87.26 | 80.64 | 80.39 | 80.71 |
| Hessian Newton | 62.40 | **78.82** | 85.94 | — | — | — | **83.09** | **81.13** | **83.45** |
| Fisher Newton | 58.80 | 78.74 | **86.10** | 53.82 | **79.24** | **87.41** | 80.70 | 80.37 | 80.45 |

training performs best if only 3 samples can be drawn. This makes sense as the estimation of Hessian or Fisher with stochastic smoothing is not good enough with too few samples, but as soon as we have at least 10 samples, it is good enough to improve performance. For stochastic smoothing of the algorithm, (i.e., stochastic smoothing applied only to the algorithm, and the gradient of the loss afterward computed using backpropagation), we can observe a very similar behavior. While estimating the Hessian is intractable in this case, we can see improvements using the Fisher for $\geq 10$ samples. For perturbed optimizers with a Fenchel-Young loss [46], we can confirm that the number

Table 4: MNIST classification learning results. The models are a 5-layer fully connected ReLU networks with 100 (M1), 400 (M2) and 1 600 (M3) neurons per layer, as well as the convolutional LeNet-5 with sigmoid activations (M4) and LeNet-5 with ReLU activations (M5). The results are averaged over 20 seeds, and significance tests between regular training and the Newton methods are conducted. A better mean is indicated by a gray bold-face number, and a significantly better result is indicated by a black bold-face number.

| | Optim. | SGD Optimizer | | | | | Adam Optimizer | | | | |
| Ep. | Method / Model | M1 | M2 | M3 | M4 | M5 | M1 | M2 | M3 | M4 | M5 |
|---|---|---|---|---|---|---|---|---|---|---|---|
| 1 | Regular | 93.06% | 94.26% | 94.77% | 10.57% | 96.25% | 94.75% | 96.36% | 95.90% | 90.60% | 97.56% |
| 1 | Newton L. (e.w. H) | 92.95% | 94.23% | 94.74% | 10.57% | 96.14% | 94.63% | 96.30% | 96.06% | 90.32% | 97.63% |
| 1 | Newton L. (H) | 93.06% | 94.28% | 94.77% | 10.57% | 96.23% | 94.75% | 96.36% | 95.91% | 90.63% | 97.63% |
| 1 | Newton L. (F) | **94.56%** | **95.47%** | **95.36%** | 10.64% | **97.77%** | 94.86% | 96.28% | 95.95% | 90.57% | 97.49% |
| 200 | Regular | 98.13% | 98.40% | 98.46% | 99.06% | 99.07% | 98.12% | 98.46% | 98.62% | 98.95% | 99.23% |
| 200 | Newton L. (e.w. H) | 98.11% | 98.39% | 98.44% | 99.02% | 99.09% | 98.16% | 98.44% | 98.54% | 98.95% | 99.22% |
| 200 | Newton L. (H) | 98.11% | 98.42% | 98.46% | 99.04% | 99.11% | 98.12% | 98.50% | 98.63% | 98.97% | 99.20% |
| 200 | Newton L. (F) | **98.22%** | **98.56%** | **98.68%** | **99.11%** | **99.23%** | 98.09% | 98.53% | 98.66% | 98.98% | 99.21% |

of samples drawn barely affects performance. By extending the formulation to also computing the Hessian of the Fenchel-Young loss, we can compute the Newton loss, and find that we achieve improvements of more than 2% in this case. Interestingly, we find that perturbed optimizers are more sample efficient but do not improve with more samples. Also, we compare stochastic smoothing of the loss (which computes a gradient) and stochastic smoothing of the algorithm (which computes a Jacobian matrix). Smoothing of the loss is more sample efficient. On the other hand, smoothing of the algorithm performs better for $\geq 10$ samples.

## 5.3 ABLATION STUDY: CLASSIFICATION

Finally, as an ablation study, we explore the utility of Newton losses for the simple case of MNIST classification with a softmax cross-entropy loss. Note that, as softmax is already a well-behaved objective, we cannot expect improvements, and the experiment is rather to demonstrate that no loss in performance is induced through the Newton losses.

To facilitate a fair and extensive comparison, we benchmark training on 5 models and with 2 optimizers: We use 5-layer fully connected ReLU networks with 100 (M1), 400 (M2) and 1 600 (M3) neurons per layer, as well as the convolutional LeNet-5 with sigmoid activations (M4) and LeNet-5 with ReLU activations (M5). Further, we use SGD and Adam as optimizers. To evaluate both early performance and full training performance, we test after 1 and 200 epochs. As computing the Hessian inverse is trivial for these settings, we also include the element-wise Hessian (e.w. H); however, as it performs (expectedly) poorly and as it would also be too expensive in the previous experiments, we did not include it for the algorithmic losses experiments in the previous sections.

We run each experiment with 20 seeds, which allows us to perform significance tests (significance level 0.05). As displayed in Table 4, we find that the element-wise Hessian (e.w. H) performs similar to regular training. Using the empirical Hessian (H) is indistinguishable from regular training, specifically in 12 out of 20 cases it is better and significantly better in one 1 out of 20 cases, which is to be expected from equal methods (on average 1/20 tests will be significant at a significance level of 0.05). Finally, we find that the Fisher-based Newton losses perform better than regular training. Specifically, with the SGD optimizer, in 9 out of 10 settings, it is significantly better and on the remaining setting, it has a higher mean. Using Adam [48], both methods perform similarly.

## 6 CONCLUSION

In this work, we proposed Newton losses, a method for combining second-order optimization of the loss function and first-order optimization of the model. We extensively benchmarked Newton losses on multiple tasks with an array of algorithmic losses. We found that Newton losses improve performance when training with non-trivial loss functions like algorithmic losses.

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

# A PROOFS

**Theorem 2** (Gradient Descent Step Equality between (2) and (3a)+(3b)). *A gradient descent step according to (2) with arbitrary step size $\eta$ coincides with two gradient descent steps according to (3a) and (3b), where the optimization over $\theta$ has a step size of $\eta$ and the optimization over $z$ has a unit step size.*

*Proof.* Let $\theta \in \Theta$ be the current parameter vector and let $\mathbf{z} = f(\mathbf{x}; \theta)$. Then the gradient descent steps according to (3a) and (3b) with step sizes 1 and $\eta > 0$ are expressed as

$$\mathbf{z} \leftarrow \mathbf{z} - \nabla_{\mathbf{z}}\ell(\mathbf{z}) = f(\mathbf{x}; \theta) - \nabla_f \ell(f(\mathbf{x}; \theta)) \tag{20}$$

$$\theta \leftarrow \theta - \eta \, \nabla_\theta \tfrac{1}{2} \|\mathbf{z} - f(\mathbf{x}; \theta)\|_2^2 = \theta - \eta \, \frac{\partial f(\mathbf{x}; \theta)}{\partial \theta} \cdot (f(\mathbf{x}; \theta) - \mathbf{z}) \,. \tag{21}$$

Combining (20) and (21) eventually leads to

$$\theta \leftarrow \theta - \eta \, \frac{\partial f(\mathbf{x}; \theta)}{\partial \theta} \cdot (f(\mathbf{x}; \theta) - f(\mathbf{x}; \theta) + \nabla_f \ell(f(\mathbf{x}; \theta))) = \theta - \eta \, \nabla_\theta \ell(f(\mathbf{x}; \theta)), \tag{22}$$

which is exactly a gradient descent step of problem (1) starting at $\theta \in \Theta$ with step size $\eta$. $\qquad \square$

**Theorem 3** (Newton Step Equality between (2) and (3a)+(3b) for $m = 1$). *In the case of $m = 1$, a Newton step according to (2) with arbitrary step size $\eta$ coincides with two Newton steps according to (3a) and (3b), where the optimization over $\theta$ has a step size of $\eta$ and the optimization over $z$ has a unit step size.*

*Proof.* Let $\theta \in \Theta$ be the current parameter vector and let $\mathbf{z} = f(\mathbf{x}; \theta)$. Then applying Newton steps according to (3a) and (3b) leads to

$$\mathbf{z} \leftarrow \mathbf{z} - (\nabla_{\mathbf{z}}^2 \ell(\mathbf{z}))^{-1} \nabla_{\mathbf{z}} \ell(\mathbf{z}) = f(\mathbf{x}; \theta) - (\nabla_f^2 \ell(f(\mathbf{x}; \theta)))^{-1} \nabla_f \ell(f(\mathbf{x}; \theta)) \tag{23}$$

$$\theta \leftarrow \theta - \eta \left( \nabla_\theta^2 \frac{1}{2} \|\mathbf{z} - f(\mathbf{x}; \theta)\|_2^2 \right)^{-1} \nabla_\theta \frac{1}{2} \|\mathbf{z} - f(\mathbf{x}; \theta)\|_2^2 \tag{24}$$

$$= \theta - \eta \left( \frac{\partial}{\partial \theta} \left[ \frac{\partial f(\mathbf{x}; \theta)}{\partial \theta} \cdot (f(\mathbf{x}; \theta) - \mathbf{z}) \right] \right)^{-1} \frac{\partial f(\mathbf{x}; \theta)}{\partial \theta} \cdot (f(\mathbf{x}; \theta) - \mathbf{z}) \tag{25}$$

$$= \theta - \eta \left( \frac{\partial}{\partial \theta} \left[ \frac{\partial f(\mathbf{x}; \theta)}{\partial \theta} \right] (f(\mathbf{x}; \theta) - \mathbf{z}) + \left( \frac{\partial f(\mathbf{x}; \theta)}{\partial \theta} \right)^2 \right)^{-1} \frac{\partial f(\mathbf{x}; \theta)}{\partial \theta} \cdot (f(\mathbf{x}; \theta) - \mathbf{z})$$

Inserting (23), we can rephrase the update above as

$$
\begin{aligned}
\theta \leftarrow \theta - \eta &\left( \frac{\partial}{\partial \theta} \left[ \frac{\partial f(\mathbf{x}; \theta)}{\partial \theta} \right] (\nabla_f^2 \ell(f(\mathbf{x}; \theta)))^{-1} \nabla_f \ell(f(\mathbf{x}; \theta)) + \left( \frac{\partial f(\mathbf{x}; \theta)}{\partial \theta} \right)^2 \right)^{-1} \\
&\cdot \frac{\partial f(\mathbf{x}; \theta)}{\partial \theta} \cdot (\nabla_f^2 \ell(f(\mathbf{x}; \theta)))^{-1} \nabla_f \ell(f(\mathbf{x}; \theta))
\end{aligned}
\tag{26}
$$

By applying the chain rule twice, we further obtain

$$
\begin{aligned}
\nabla_\theta^2 \ell(f(\mathbf{x}; \theta)) &= \frac{\partial}{\partial \theta} \left[ \frac{\partial f(\mathbf{x}; \theta)}{\partial \theta} \nabla_f \ell(f(\mathbf{x}; \theta)) \right] \\
&= \frac{\partial}{\partial \theta} \left[ \frac{\partial f(\mathbf{x}; \theta)}{\partial \theta} \right] \nabla_f \ell(f(\mathbf{x}; \theta)) + \frac{\partial f(\mathbf{x}; \theta)}{\partial \theta} \frac{\partial}{\partial \theta} \nabla_f \ell(f(\mathbf{x}; \theta)) \\
&= \frac{\partial}{\partial \theta} \left[ \frac{\partial f(\mathbf{x}; \theta)}{\partial \theta} \right] \nabla_f \ell(f(\mathbf{x}; \theta)) + \frac{\partial f(\mathbf{x}; \theta)}{\partial \theta} \nabla_f \frac{\partial}{\partial \theta} \ell(f(\mathbf{x}; \theta)) \\
&= \frac{\partial}{\partial \theta} \left[ \frac{\partial f(\mathbf{x}; \theta)}{\partial \theta} \right] \nabla_f \ell(f(\mathbf{x}; \theta)) + \left( \frac{\partial f(\mathbf{x}; \theta)}{\partial \theta} \right)^2 \nabla_f^2 \ell(f(\mathbf{x}; \theta)),
\end{aligned}
$$

which allows us to rewrite (26) as

$$
\begin{aligned}
\theta' &= \theta - \left( (\nabla_f^2 \ell(f(\mathbf{x}; \theta)))^{-1} \nabla_\theta^2 \ell(f(\mathbf{x}; \theta)) \right)^{-1} (\nabla_f^2 \ell(f(\mathbf{x}; \theta)))^{-1} \nabla_\theta \ell(f(\mathbf{x}; \theta)) \\
&= \theta - (\nabla_\theta^2 \ell(f(\mathbf{x}; \theta)))^{-1} \nabla_\theta \ell(f(\mathbf{x}; \theta)),
\end{aligned}
$$

which is exactly a single Newton step of problem (1) starting at $\theta \in \Theta$. $\qquad \square$

## B  FURTHER EXAMPLES FOR NEWTON LOSSES

A less trivial example is the binary cross-entropy (BCE) loss with

$$\ell_{\text{BCE}}(y) = \text{BCE}(y, p) = -\sum_{i=1}^{m} p_i \log y_i + (1 - p_i) \log(1 - y_i), \tag{27}$$

where $p \in \Delta_m$ is a probability vector encoding of the ground truth.

**Example 3** (Binary cross-entropy loss)**.** *For the BCE loss, the induced Newton loss is given as*

$$\ell_{\text{BCE}}^*(y) = \frac{1}{2} \|z^\star - y\|_2^2, \tag{28}$$

*where the element-wise Hessian variant is*

$$z_E^\star = -\operatorname{diag}\left(-p \oslash y^2 + (1 - p) \oslash (1 - y)^2\right)^{-1} (p \oslash y - (1 - p) \oslash (1 - y)) + y, \tag{29}$$

*the empirical Hessian variant is*

$$z_H^\star = -\operatorname{diag}\left(\mathbb{E}_y\left[-p \oslash y^2 + (1 - p) \oslash (1 - y)^2\right]\right)^{-1} (p \oslash y - (1 - p) \oslash (1 - y)) + y, \tag{30}$$

*and the empirical Fisher variant is*

$$z_F^\star = -\mathbb{E}_y\left[(p \oslash y - (1 - p) \oslash (1 - y))(p \oslash y - (1 - p) \oslash (1 - y))^\top\right]^{-1} \tag{31}$$
$$(p \oslash y - (1 - p) \oslash (1 - y)) + y,$$

*and where $\odot$ and $\oslash$ are element-wise operations.*

The BCE loss is often extended using the logistic sigmoid function to what is called the sigmoid binary cross-entropy loss (SBCE), defined as

$$\ell_{\text{SBCE}}(y) = \text{BCE}(\sigma(y), p) \qquad \text{where} \qquad \sigma(x) = \frac{1}{1 + \exp(-x)}. \tag{32}$$

**Example 4** (Sigmoid Binary cross-entropy loss)**.** *For the SBCE loss, the induced Newton loss is given as*

$$\ell_{\text{SBCE}}^*(y) = \frac{1}{2} \|z^\star - y\|_2^2 \tag{33}$$

*where the element-wise Hessian variant is*

$$z_E^\star = -\operatorname{diag}\left(\sigma(y) - \sigma(y)^2\right)^{-1} (\sigma(y) - p) + y, \tag{34}$$

*the empirical Hessian variant is*

$$z_H^\star = -\operatorname{diag}\left(\mathbb{E}_y\left[\sigma(y) - \sigma(y)^2\right]\right)^{-1} (\sigma(y) - p) + y, \tag{35}$$

*and the empirical Fisher variant is*

$$z_F^\star = -\mathbb{E}_y\left[(\sigma(y) - p)(\sigma(y) - p)^\top\right]^{-1} (\sigma(y) - p) + y, \tag{36}$$

*and where $\odot$ and $\oslash$ are element-wise operations.*

## C    REGULARIZERS $\Omega$

In this section, we provide a more detailed discussion, how the regularization term $\Omega$ in (4) induces different iterative optimization methods.

The simple setting, where (3a) represents a basic gradient descent, i.e.,

$$\mathbf{z}^\star \leftarrow \mathbf{z} - \eta \nabla \ell(\mathbf{z}), \quad \mathbf{z} = f(\mathbf{x}; \theta), \tag{37}$$

can be obtained by choosing a regularization term as

$$\Omega(\mathbf{z}, f(\mathbf{x}; \theta)) = \ell(f(\mathbf{x}; \theta)) + \eta \nabla_f \ell(f(\mathbf{x}; \theta))^\top (\mathbf{z} - f(\mathbf{x}; \theta)) - \ell(\mathbf{z}) \tag{38}$$

$$+ \frac{1}{2} (\mathbf{z} - f(\mathbf{x}; \theta))^\top (\mathbf{z} - f(\mathbf{x}; \theta)). \tag{39}$$

Then, the first-order optimality conditions for $\min_{\mathbf{z}} \ell(\mathbf{z}) + \Omega(\mathbf{z}, f(\mathbf{x}; \theta))$ are

$$\nabla_z (\ell(\mathbf{z}) + \Omega(\mathbf{z}, f(\mathbf{x}; \theta))) = \eta \nabla_f \ell(f(\mathbf{x}; \theta)) + \mathbf{z} - f(\mathbf{x}; \theta) = 0, \tag{40}$$

which leads to the gradient step $\mathbf{z} = f(\mathbf{x}; \theta) - \eta \nabla_f \ell(f(\mathbf{x}; \theta))$.

Alternatively, we can ask for the choice of the regularization term $\Omega$ corresponding to a Newton step in (3a), i.e.,

$$\mathbf{z}^\star \leftarrow \mathbf{z} - \eta (\nabla^2 \ell(\mathbf{z}))^{-1} \nabla \ell(\mathbf{z}), \quad \mathbf{z} = f(\mathbf{x}; \theta). \tag{41}$$

In this case, by choosing

$$\Omega(\mathbf{z}, f(\mathbf{x}; \theta)) = \ell(f(\mathbf{x}; \theta)) + \eta (\mathbf{z} - f(\mathbf{x}; \theta))^\top \nabla_f \ell(f(\mathbf{x}; \theta)) \tag{42}$$

$$+ \frac{1}{2} (\mathbf{z} - f(\mathbf{x}; \theta))^\top \nabla_f^2 \ell(f(\mathbf{x}; \theta))(\mathbf{z} - f(\mathbf{x}; \theta)) - \ell(\mathbf{z}), \tag{43}$$

the first-order optimality conditions for $\min_{\mathbf{z}} \ell(\mathbf{z}) + \Omega(\mathbf{z}, f(\mathbf{x}; \theta))$ are

$$\nabla_z (\ell(\mathbf{z}) + \Omega(\mathbf{z}, f(\mathbf{x}; \theta))) = \eta \nabla_f \ell(f(\mathbf{x}; \theta)) + \nabla_f^2 \ell(f(\mathbf{x}; \theta))(\mathbf{z} - f(\mathbf{x}; \theta)) = 0, \tag{44}$$

which is equivalent to a Newton step $\mathbf{z} = f(\mathbf{x}; \theta) - \eta (\nabla^2 \ell(f(\mathbf{x}; \theta)))^{-1} \nabla \ell(f(\mathbf{x}; \theta))$.

## D   RUNTIMES

In this supplementary material, we provide and discuss runtimes for the experiments.

In the differentiable sorting and ranking experiment, as shown in Table 5, we observe that the runtime from regular training compared to the Newton Loss with the Fisher is only marginally increased. This is because computing the Fisher and inverting it is very inexpensive. We observe that the Newton loss with the Hessian, however, is more expensive: due to the implementation of the differentiable sorting and ranking operators, we compute the Hessian by differentiating each element of the gradient, which makes this process fairly expensive. An improved implementation could make this process much faster. Nevertheless, there is always some overhead to computing the Hessian compared to the Fisher.

Table 5:   Runtimes for the differentiable sorting results corresponding to Table 1. Times of full training in seconds.

|  | $n = 5$ | | | $n = 10$ | | |
|---|---|---|---|---|---|---|
|  | DSN | NeuralSort | SoftSort | DSN | NeuralSort | SoftSort |
| Regular | 4214 | 3712 | 3683 | 6192 | 5236 | 5011 |
| NL (Hessian) | 8681 | 4057 | 4216 | 22617 | 6154 | 6010 |
| NL (Fisher) | 4232 | 3762 | 3730 | 6239 | 5153 | 5098 |

In Table 6, we show the runtimes for the shortest-path experiment with AlgoVision. Here, we observe that the runtime overhead is very small.

Table 6:   Runtimes for the shortest-path results corresponding to Table 2. Times of full training in seconds.

| Algorithm Loop Loss | For | | While | |
|---|---|---|---|---|
|  | $L_1$ | $L_2^2$ | $L_1$ | $L_2^2$ |
| Regular | 624 | 606 | 911 | 913 |
| Fisher Newton | 608 | 634 | 920 | 917 |

In Table 7, we show the runtimes for the shortest-path experiment with stochastic methods. Here, we observe that the runtime overhead is also very small. Here, the Hessian is also cheap to compute as it is not computed with automatic differentiation.

Table 7:   Runtimes for the shortest-path results corresponding to Table 3. Times of full training in seconds.

| Loss # Samples | SS of loss | | | SS of algorithm | | | PO w/ FY loss | | |
|---|---|---|---|---|---|---|---|---|---|
|  | 3 | 10 | 30 | 3 | 10 | 30 | 3 | 10 | 30 |
| Regular | 894 | 1419 | 3183 | 916 | 1401 | 3170 | 646 | 1157 | 2947 |
| Hessian Newton | 902 | 1398 | 3159 | — | — | — | 666 | 1161 | 2971 |
| Fisher Newton | 882 | 1394 | 3217 | 901 | 1382 | 3200 | 667 | 1168 | 2973 |

# E    CONVERGENCE PLOTS FOR THE CLASSIFICATION EXPERIMENT

In this Figure 1, we provide convergence plots for the classification experiment. Here, we consider model M3, and the MNIST as well as the CIFAR-10 data sets. We show the convergence over the first 50 epochs.

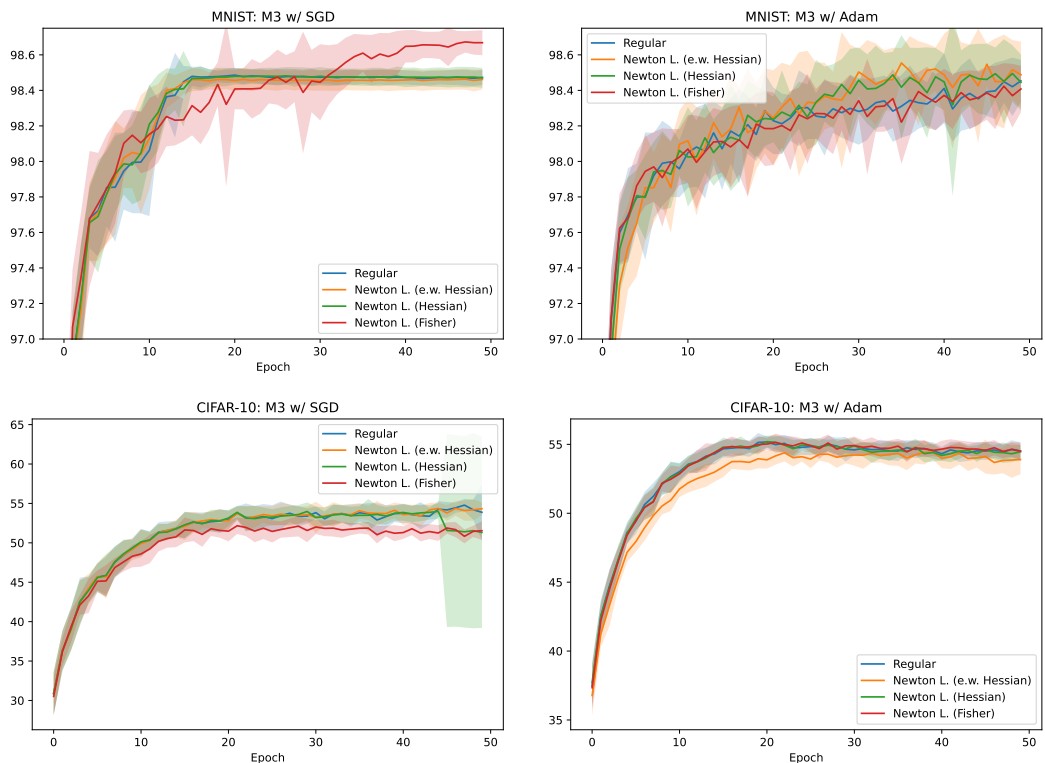

Figure 1:  Convergence plots of the test accuracy for the classification experiment on MNIST (top) and CIFAR-10 (bottom) for model M3. On the left, we use the SGD optimizer and on the right, we use the Adam optimizer. Results are averaged over 20 seeds and the standard deviations are marked in shaded color. We note that for CIFAR-10 / SGD (bottom left) one of the seeds for "Newton L. (Hessian)" crashed numerically, which increased the standard deviation at this point.

