# OpenReview forum: "Newton Losses: Efficiently Including Second-Order Information into Gradient Descent"
_ICLR.cc/2023/Conference — Submitted to ICLR 2023_

### Official Review · Reviewer_T2yc · 2022-10-22

**Confidence:** 4
**Correctness:** 4
**Technical Novelty And Significance:** 2
**Empirical Novelty And Significance:** 2
**Recommendation:** 3

**Clarity, Quality, Novelty And Reproducibility:**

There are a few unclear definitions and notations:
1) In Eq. (4), $\Omega$ is used to denote the regularizer both for the model parameters and outputs. This is confusing since these two are of different natures.
2) Page 4: what does element-wise Hessian mean? Does that mean diagonal approximation?
3) I could not find the definition of $\hat{y}$. Also, when defining expectations, please be more clear about what the expectation is wrt. For instance, Remark 1: the empirical Fisher should be an expectation wrt the training labels.

**Strength And Weaknesses:**

### Strengths:
* The motivation for the approach is clear.

### Weaknesses:
* The authors have missed important previous work that introduces such techniques. Namely, lifted networks [1] propose a two-stage optimization scheme by introducing latent variables and matching the output of each later with the latent variables. More importantly, LocoProp [2] introduced the idea of setting layerwise targets and iteratively minimizing layerwise losses to match the targets. The method proposed in this paper is a special case of LocoProp where 1) only a single target is formed for the last layer via a Newton step (this was already discussed in LocoProp in terms of a natural gradient descent (NGD) targets, and the authors show that the final update wrt to the parameters corresponds to NGD), 2) the loss is set to be the squared loss whereas LocoProp uses more advanced Bregman divergences, 3) the model parameters are updated using a single step whereas, in LocoProp, the authors try multiple steps on the fixed point problem and show that the final update on the parameters corresponds to a preconditioned update.

[1] Miguel Carreira-Perpinan, and Weiran Wang. "Distributed optimization of deeply nested systems." In Artificial Intelligence and Statistics, pp. 10-19. PMLR, 2014.
[2] Ehsan Amid, Rohan Anil, and Manfred Warmuth. "Locoprop: Enhancing Backprop via local loss optimization." In International Conference on Artificial Intelligence and Statistics, pp. 9626-9642. PMLR, 2022.


**Summary Of The Paper:**

The authors propose Newton losses, which breaks the loss optimization into two stages. The latent variable $z$ behaves as a target for the model output and is formed using a Newton step on the loss wrt the activations. The parameters are then optimized to match the target by minimizing a squared loss. The authors provide examples of different losses and validate the results experimentally.


**Summary Of The Review:**

The authors need to distinguish their work from lifted networks and LocoProp. Currently, their proposed method is a special case of LocoProp on the final layer with a single local iteration.

---

> ### Author Response · Authors · 2022-11-19
> **Author response.**
>
> We thank the reviewer T2yc for the review and appreciate that the reviewer finds the motivation for our approach clear.
>
> > Lifted networks [1] and LocoProp [2]
>
> Thank you for bringing these references to our attention. We will include a detailed discussion of them in the camera-ready version of the paper (as we do not have additional space for the current revision). In the following, we distinguish our work from these works.
>
> **LocoProp**
> There are a few decisive differences between our method and LocoProp.
> Specifically, to go from LocoProp to our method, one would have to (1) apply it only to the outputs of the neural network, (2) use different update schemes for different parts of the neural network (which is neither mentioned nor done in [2]), (3) update model parameters using a single step.
> Apart from these methodological differences, the goal and approach in our work are different from those of LocoProp:
> In LocoProp, the goal is to operate on simple standard losses like squared losses and cross entropy.
> We show that our method does not improve performance for the case of these losses (for squared loss we show theoretically that it's Newton loss is exactly the same, and for cross-entropy we demonstrate empirically that, while it works, it does not have a benefit).
> In contrast, our method is designed for application to highly non-convex and non-trivial losses like algorithmic sorting and ranking losses.
> For these losses, our method improves performance substantially.
> Such losses have not been considered in the LocoProp paper, and we would assume that LocoProp would not provide a special advantage on these losses.
>
> **Lifted Networks**
> As for lifted networks [1], the differences are that (1) like for LocoProp only squared error and cross-entropy are considered as loss functions, (2) no second order optimization is attempted for lifted networks, (3) the core idea of lifted networks is to introduce the activations of the neurons in the hidden layers as auxiliary variables, which are fixed in an alternating optimization scheme when optimizing the weights, while in the second step of the alternating optimization the weights are fixed and the auxiliary variables are optimized, posing substantial differences.
> We note that the "alternating" in their optimization scheme is alternating in a different way from how our method is alternating.
>
> > In Eq. (4), $\Omega$ is used to denote the regularizer both for the model parameters and outputs. This is confusing since these two are of different natures.
>
> We understand that this could lead to confusions and propose to distinguish them into $\Omega_\theta$ and $\Omega_\mathbf{z}$.
>
> > Page 4: what does element-wise Hessian mean? Does that mean diagonal approximation?
>
> As shown in Definition 1, element-wise Hessian refers to the Hessian of a single sample, i.e., not the population Hessian.
>
> > I could not find the definition of $\hat{y}$.
>
> We do not use any $\hat{y}$ and therefore do not understand this comment.
>
> > Also, when defining expectations, please be more clear about what the expectation is wrt. For instance, Remark 1: the empirical Fisher should be an expectation wrt the training labels.
>
> Thanks, we clarified it in the revision.

---

> > ### Comment · Reviewer_T2yc · 2022-11-19
> > **Thank you**
> >
> > Thank you for your response.
> >
> > -  I encourage the authors to carefully check the lifted networks and LocoProp references. Although LocoProp discusses the loss construction per layer, you can also devise losses per sub-block of the network, which in this case is the entire network (i.e., only construct a loss in the last layer). The experiments in the LocoProp paper focus on multiple iterations per forward-backward pass, but the formulation is more generic and includes Newton losses as a special case. To be precise, when 1) the target in the last layer is formed using a Newton step on the current post activations, 2) the loss is set to squared loss, and 3) the number of local iterations is set to one.
> >
> > - Also I was referring to $\bar{y}$ (not $\hat{y}$) which is not specified clearly.
> >
> > I tend to keep my score.

---

> > > ### Author Response · Authors · 2022-11-20
> > > **Response**
> > >
> > > Thank you for your quick reply!
> > >
> > > > I encourage the authors to carefully check the lifted networks and LocoProp references. Although LocoProp discusses the loss construction per layer, you can also devise losses per sub-block of the network, which in this case is the entire network (i.e., only construct a loss in the last layer). The experiments in the LocoProp paper focus on multiple iterations per forward-backward pass, but the formulation is more generic and includes Newton losses as a special case. To be precise, when 1) the target in the last layer is formed using a Newton step on the current post activations, 2) the loss is set to squared loss, and 3) the number of local iterations is set to one.
> > >
> > > You are right that, when LocoProp is modified by 1), 2) and 3) as you sketched above, the formulation becomes equivalent to Newton losses. However, importantly, when modifying vanilla gradient descent or standard backpropagation by 1) and 2), the formulation also becomes equivalent to Newton losses.
> > > Further, we would like to note that, while LocoProp discusses both first and second-order approaches, a combination of both approaches in the same learning framework is neither considered nor discussed.
> > > Our approach combines first-order and second-order optimization methods in a way that exploits the underlying problem structure of having a hard-to-optimize loss.
> > > Accordingly, also the difference in the applications and experiments (we use algorithmic losses, LocoProp etc. use regression and classification losses) matters a lot.
> > >
> > > > Also I was referring to $\bar y$ (not $\hat y$) which is not specified clearly.
> > >
> > > We respectfully disagree. Throughout our paper, at each use of $\bar{y}$, we specify $\bar y = f(x;\theta)$. If you should have found any occurrence where we did not specify $\bar{y}$, please specify where it is.

---

### Official Review · Reviewer_qAQi · 2022-10-24

**Confidence:** 3
**Correctness:** 2
**Technical Novelty And Significance:** 2
**Empirical Novelty And Significance:** 1
**Recommendation:** 5

**Clarity, Quality, Novelty And Reproducibility:**

I generally found the clarity low throughout, though this could be in-part due to my unfamiliarity with some of the problem settings. For instance, I've read the introduction to the experiments section several times and still have no idea what an "algorithmic supervision" task is.

The main source of confusion was the math though; much of the math very hand-wavy and imprecise to me, as mentioned above.

The approach seems somewhat novel, and the experimental results suggest that the newton loss can sometimes lead to strong performance improvements over using the regular loss function.

The empirical results serve mostly as a sanity check and don't seem to reveal any new insights in their own right. As far as reproducibility is concerned, I'm not confident I could reproduce these results unless a code release is planned.

**Strength And Weaknesses:**

**Strengths** Breaking the update into an update wrt the loss function and an update wrt the model seems like an interesting and practical way to incorporate second-order information without computing the hessian wrt the model.

**Weaknesses** The main weakness is that the math felt very imprecise to me throughout. For instance, throughout the paper argmin's are computed by arguing that the gradient is 0 at the argmin, but this isn't necessarily true since that point might have been outside of the domain. Similarly, there's often an implicit assumption that the model's range covers the entire domain, since this is how we get equivalence of $f(x;\theta)=z^*\in\text{argmin}_z\ell(z)+\Omega(z,f(x;\theta))$, which is used in the proof of Lemma 1 showing that the iterative and two-step method converge to the same points. This is not necessarily true without additional assumptions on the model class.

Section 4 cataloguing various algorithmic supervision losses seemed mostly unnecessary and would potentially work better as an appendix

**Summary Of The Paper:**

The paper introduces an approach to efficiently computing updates that incorporate second-order information. The idea is to decompose the update into two parts: a second-order update of the loss function, and a first-order update of the model. Notably, this avoids having to compute the hessian wrt the model parameters, which would be expensive for complex models like neural networks. It's shown that the two-step scheme can be equivalent to gradient descent and to the newton step algorithm for certain choices of the updates. The approach is tested empirically on algorithmic supervision tasks.

**Summary Of The Review:**

The approach seems interesting and reasonably motivated, but I am not confident that the math actually checks out and would be hesitant to recommend acceptance.

---

> ### Author Response · Authors · 2022-11-19
> **Author response.**
>
> We thank reviewer qAQi for the feedback and appreciate that the reviewer finds our method interesting and practical.
>
> > Breaking the update into an update wrt the loss function and an update wrt the model seems like an interesting and practical way to incorporate second-order information without computing the hessian wrt the model.
>
> Thank you for appreciating that our method is interesting and a practical way of solving the problem at hand.
>
> > The main weakness is that the math felt very imprecise to me throughout. For instance, throughout the paper argmin's are computed by arguing that the gradient is 0 at the argmin, but this isn't necessarily true since that point might have been outside of the domain. Similarly, there's often an implicit assumption that the model's range covers the entire domain, since this is how we get equivalence of $f(x;\theta)=z^*\in\arg\min_z \ell(z) + \Omega(z,f(x;\theta))$, which is used in the proof of Lemma 1 showing that the iterative and two-step method converge to the same points. This is not necessarily true without additional assumptions on the model class.
>
> It is correct that the gradient need not be zero at the argmin if domain is restricted; however, we are assuming a real valued space of outputs as well as a general neural network, which does not have such restrictions. Hence, the gradient vanishes at the argmin (since no domain boundary problems are to be expected). We note that this assumption equally also applies for Newton's method.
> Further, we would like to argue that the model's range covering the entire domain is a fairly standard assumption, which we also state in our paper. We propose to add an additional clarification to the camera-ready version of the paper.
>
> > Section 4 cataloguing various algorithmic supervision losses seemed mostly unnecessary and would potentially work better as an appendix
>
> We agree that one could argue placing Section 4 in the appendix; however, we want to maintain readability for people unfamiliar with the broad literature of algorithmic supervision losses, which is why we prefer to keep these details in the paper.
>
> > I generally found the clarity low throughout, though this could be in-part due to my unfamiliarity with some of the problem settings. For instance, I've read the introduction to the experiments section several times and still have no idea what an "algorithmic supervision" task is.
>
> We discuss algorithmic supervision in Section 4, specifically, we clarify that algorithmic supervision refers to "problems where an algorithm is applied to the predictions of a model and only the outputs of the algorithm are supervised", which also matches the definition by Petersen et al., 2021 [18 in the paper].
> We propose to extend the discussion in Section 4 for the camera-ready paper.
>
> > The main source of confusion was the math though; much of the math very hand-wavy and imprecise to me, as mentioned above.
>
> Could you please make some concrete suggestions or indicate what was "hand-wavy".
>
> > The approach seems somewhat novel, and the experimental results suggest that the newton loss can sometimes lead to strong performance improvements over using the regular loss function.
>
> Many thanks for the appreciation of our numerical work.
>
> > The empirical results serve mostly as a sanity check and don't seem to reveal any new insights in their own right. As far as reproducibility is concerned, I'm not confident I could reproduce these results unless a code release is planned.
>
> We will make the code available once the paper is accepted.

---

### Official Review · Reviewer_fJV7 · 2022-10-24

**Confidence:** 2
**Correctness:** 2
**Technical Novelty And Significance:** 1
**Empirical Novelty And Significance:** 2
**Recommendation:** 3

**Clarity, Quality, Novelty And Reproducibility:**

**Clarity** I found it extremely difficult to follow the paper and it is not clear to me where the line between background and newly introduced material is.

**Quality** Difficult to judge due to the lack of clarity.

**Novelty** See quality.

**Strength And Weaknesses:**

Strengths:
* The performance improvements are consistent across the experiments.

Weaknesses:
* I find it extremely hard to understand what the contribution of the paper is supposed to be. Is it simply to use second order optimization on the the inner loss in bi-level optimization? Is it formulating some algorithmic losses as bi-level optimization problems? I also don't understand why the introduction opens up with a discussion of second-order optimization for neural networks, this does not seem relevant at all to what (I think) the paper is doing.
* Related, I found it difficult to read the paper. The paper seems relatively practical to me, so I don't quite understand the choice to opt for a formal/mathematical style (Definition, Lemma, Theorem, Remark). Section 2 in particular seemed extremely verbose to me, while section 4 is lacking a lot of detail (without giving the reader a strong sense for what is going on in terms of the big picture).
* The positioning is similarly opaque to me. While the related work cites a bunch of papers, I did not find a clear statement as to what gap in the literature the paper is trying to close and where it is improving over prior works.

Minor comments:
* Based on the title, I would have expected the paper to propose a method that includes second order information in the objective of general objective functions (similar to e.g. sharpness-aware minimization). I would suggest altering the title to make it more descriptive of the contents.
* Inlining Tabs 2 & 3 makes reading the corresponding sections unnecessarily difficult.

**Summary Of The Paper:**

The paper proposes to formulate some algorithmic losses (e.g. sorting) as bi-level optimization problems and solve the inner problem with second-order optimization. It reports performance gains over regular optimization

**Summary Of The Review:**

The contribution of the paper is not clear to me, so I'm arguing for rejection.

---

> ### Author Response · Authors · 2022-11-19
> **Author response.**
>
> We thank reviewer fJV7 for the feedback and appreciate that the reviewer appreciates our consistent performance improvements across the experiments.
>
> > The performance improvements are consistent across the experiments.
>
> Thank you for acknowledging that that the performance improvements we obtain are consistent across the experiments.
>
> > Based on the title, I would have expected the paper to propose a method that includes second order information in the objective of general objective functions (similar to e.g. sharpness-aware minimization). I would suggest altering the title to make it more descriptive of the contents.
>
> We believe that our title is fitting as we indeed do "propose a method that includes second order information in the objective of general objective functions". Therefore, we do not understand why the title seems inappropriate to you.
>
> > I find it extremely hard to understand what the contribution of the paper is supposed to be. Is it simply to use second order optimization on the the inner loss in bi-level optimization? Is it formulating some algorithmic losses as bi-level optimization problems? I also don't understand why the introduction opens up with a discussion of second-order optimization for neural networks, this does not seem relevant at all to what (I think) the paper is doing.
>
> The main contribution of our work is the introduction of a training scheme for deep neural networks that combines first-order and second-order optimization methods in a way that exploits the underlying problem structure of having a hard-to-optimize loss. As the proposed approach relies on the second-order component, discussing the state-of-the-art of second-order approaches for neural network training seems appropriate to us.
>
> > Related, I found it difficult to read the paper. The paper seems relatively practical to me, so I don't quite understand the choice to opt for a formal/mathematical style (Definition, Lemma, Theorem, Remark). Section 2 in particular seemed extremely verbose to me, while section 4 is lacking a lot of detail (without giving the reader a strong sense for what is going on in terms of the big picture).
>
> As our presented method is a very general approach, and not a mere technical heuristic with limited applicability, we tried to justify it mathematically as rigorous as possible -- hence the formal style.
> We would like to note that Section 4 is a review of applications for our method in the context of our method. For complete details on each of these applications, we refer to the respective related work.
>
> > The positioning is similarly opaque to me. While the related work cites a bunch of papers, I did not find a clear statement as to what gap in the literature the paper is trying to close and where it is improving over prior works.
>
> The gap in the literature is that there are first-order methods and second-order methods, and we propose a method that can be seen as in between.
> In the domain of algorithmic losses, there is the overall goal to make these losses more effective, which our method does.
> We propose to add a respective explicit statement in the camera-ready version.

---

### Official Review · Reviewer_GdUr · 2022-10-24

**Confidence:** 5
**Clarity, Quality, Novelty And Reproducibility:** Clarity is fair. Novelty is below the…
**Correctness:** 3
**Technical Novelty And Significance:** 1
**Empirical Novelty And Significance:** Not applicable
**Recommendation:** 3

**Strength And Weaknesses:**

I don't see a strength in the formulation. Personally I have tried this method without introducing the L2 term and my finding is not quite satisfactory. Generally speaking when the Newton's method is only applied on the loss function it is a pretty weak method. Another way to think about it is if the loss function is a pure L2 loss, say for regression, then first order and second order methods are the same.

Also, if the authors are formulating the problem in a usable way, the L2 term should not be necessary. The second order information of the loss should be the only necessary thing to improve the training.

Lacking larger scale comparison is also a main problem with the paper. It is hard to say a piece of work is useful without scaling to imagenet-like datasets.


**Summary Of The Paper:**

The authors propose to use second order method only on the loss function -- not throughout the whole network in the training process.
They are also putting a L2 term in the formulation.

**Summary Of The Review:**

Based on the limited novelty I vote for rejection.

---

> ### Author Response · Authors · 2022-11-19
> **Author response.**
>
> We thank reviewer GdUr for the feedback.
>
> > I don't see a strength in the formulation. Personally I have tried this method without introducing the L2 term and my finding is not quite satisfactory. Generally speaking when the Newton's method is only applied on the loss function it is a pretty weak method. Another way to think about it is if the loss function is a pure L2 loss, say for regression, then first order and second order methods are the same.
>
> You are right that if the original loss function is a pure L2 loss, then first order and second order methods are the same, as we already discuss in Example 1. However, we would like to point out that our method is only intended for cases where the loss function is hard-to-optimize and non-convex such as algorithmic losses, i.e., it is not intended for typical classification and regression losses. We show that our method (as expected) does not have an advantage in the case of typical classification losses. However, on hard-to-optimize and non-convex, we do observe substantial performance improvements.
> As even reviewer fJV7 acknowledges, the performance improvements we find are consistent across experiments. Without precise information about what you tried, it is difficult to know why your approach was unsatisfactory, while ours leads to improvements. Not using the L2 term and/or using a standard classification loss like cross-entropy could explain the discrepancy to your observations.
>
> > Also, if the authors are formulating the problem in a usable way, the L2 term should not be necessary. The second order information of the loss should be the only necessary thing to improve the training.
>
> It is not quite clear to us what is meant by "if the authors are formulating the problem in a usable way, the L2 term should not be necessary."
> Could you please clarify what you mean with "formulating the problem in a usable way" and indicate why the L2 term would not be necessary?
>
> > Lacking larger scale comparison is also a main problem with the paper. It is hard to say a piece of work is useful without scaling to imagenet-like datasets.
>
> As for larger scale comparisons, we respectfully disagree. Although imagenet-like data sets are certainly an important application setting, there are other scenarios where an approach like ours can lead to valuable gains. An important point that we are raising in this respect are algorithmic supervision tasks, which are characterized by the special difficulty that the loss is not decomposable (i.e., not composed of individual losses per training example) and non-convex. For such cases our methods opens up the possibility to use second order information.

---

> > ### Comment · Reviewer_GdUr · 2022-11-20
> > **extra comments**
> >
> > 1. 3(b) seems redundant in the formulation.
> >
> > 2. It is hard to publish a paper if the proposed method cannot be shown to work for simple losses. I personally have tried to use newton's method formulation to modify the gradient computation for the cross entropy loss. The intuition is to make it almost as simple to optimize as the MSE loss. It did not work well. You may see tiny differences here and there, but it is hard to show the tweak is general.

---

> > > ### Author Response · Authors · 2022-11-21
> > > **Response to extra comments.**
> > >
> > > Thanks for the reply.
> > >
> > > 1. 3(b) is integral to the formulation. Could you please clarify why you think that 3(b) is redundant? We are aware that we could (informally speaking) "overwrite the gradient computation with the update induced by the Newton step" but this is quite informal and we chose to introduce a loss to keep it more general and to allow for formal presentation and analysis of the method.
> > >
> > > 2. The problem that the paper considers are non-convex and hard-to-optimize losses. In the ablation study (Section 5.3), we consider classification with cross entropy. The cross entropy loss is a convex loss and accordingly trivial to optimize, which is why Newton losses cannot deliver any advantage as both our experiments as well as your past experience shows. For non-convex losses, however, our method achieves substantial performance gains.

---

> > > > ### Comment · Reviewer_GdUr · 2022-11-21
> > > > **n/a**
> > > >
> > > > 1. Right. My guess is with 3b the effect seems to be doubling the effect of gradient descent.

---

### Decision · Program_Chairs · 2023-01-20

**Decision:**

Reject

**Justification For Why Not Higher Score:**

Theoretically wrong analysis.

**Justification For Why Not Lower Score:**

N/A

**Metareview: Summary, Strengths And Weaknesses:**

Overall, this paper has significant technical issues: Lemma 1 and the corresponding proof are wrong. The proof of $A\subset B$ is almost like saying "obvious we are right" without any analysis. In fact, unless much stronger assumptions are provided, the set $B$ can easily be empty if the minimizer of $\ell$ and $\ell\circ f$ does not match each other.

One can consider a trivial problem where $f (x; \theta) = \theta^2$，$\ell(z) = z$, $\Omega (\theta, \theta') = 0.5\|\theta-\theta'\|^2$, $\Omega(z,z') = 0.5\|z-z'\|^2$. Suppose we only have 1 data $x$ and the output is $\theta^2$ is a scalar. Then the problem will be   $min_\theta \theta^2   ==> \theta*=0$. Now consider the set $A$:  $\theta/3 = (argmin_{\theta'} \|\theta'\|^2 + 0.5\|\theta-\theta'\|^2) = \theta$ ==> $A = 0 $ is the correct set.  However, the set $B$ is always empty. That is, $(argmin_z  z + 0.5\|z-\theta^2\|^2) = \theta^2 - 1$.  Therefore, $\theta^2 = f (x; \theta) = (argmin_z  z + 0.5\|z-\theta^2\|^2) = \theta^2-1$ will never hold. $B = \emptyset$.

If one thinks this example is too trivial, let us consider a simple regression problem $f (x; \theta) = x'\theta$. And let the loss be a simple MSE loss, with observation $Y$ encoded in the loss function:  $\min_\theta \|\|X\theta - Y\|\|^2$. Now suppose the observation has noise so that the optimal training loss is NOT 0, which is common. Then the set $B$ will be $argmin_z \|z - Y\|^2 + 0.5\|z-X\theta\|^2 = 2Y/3 + X\theta/3$. By requirement of B, we need $2Y/3 + X\theta/3 = f (X; \theta) = X\theta ==> Y = X\theta$. This is impossible since the optimal training loss > 0 in the noisy setting.

Overall, unless extra assumptions on the losses function and the dataset itself are made, this key lemma of the paper is not correct.


**Summary Of Ac-Reviewer Meeting:**

N/A